# Unraveling the liquid gliding on vibrating solid liquid interfaces with dynamic nanoslip enactment

Amir Farokh Payam [1,2] ✉, Bogyoung Kim[3], Doojin Lee [3] ✉ &
Nikhil Bhalla [1,2] ✉

Slip length describes the classical no-slip boundary condition violation of Newtonian fluid mechanics, where fluids glide on the solid surfaces. Here, we propose a new analytical model validated by experiments for characterization of the liquid slip using vibrating solid surfaces. Essentially, we use a microfluidic system integrated with quartz crystal microbalance (QCM) to investigate the relationship between the slip and the mechanical response of a vibrating solid for a moving fluid. We discover a liquid slip that emerges especially at high flow rates, which is independent of the surface wetting condition, having significant contributions to the changes in resonant frequency of the vibrating solid and energy dissipation on its surface. Overall, our work will lead to consideration of 'missing slip' in the vibrating solid-liquid systems such as the QCM-based biosensing where traditionally frequency changes are interpreted exclusively with mass change on the sensor surface, irrespective of the flow conditions.

The dynamics of fluid flow at nano/micro scale plays a key role in numerous vital areas including high resolution two/three-dimensional printing[1], biology[2,3], micro/nanofluidics[4,5] and precise elucidation of transport phenomena at small scales[6–8]. This has raised the need for understanding the physics of flow at the interfaces between the fluid and solid surfaces, especially to know whether no-slip boundary condition (BC) at the interface is still valid for micro/nanoscale fluidic devices and sensors[9–12]. It has been previously reported that the classical no-slip boundary condition does not always apply at small scales since boundary slip effect arises more significantly when the system size decreases below sub-micrometer[13]. Indeed, the implementation of proper BC has been a debating issue ever since microfluidic devices were first developed, whilst the driving force of micro/nano scale fluidic systems has diversified with the invention of new methods for fluid propulsion[14,15].

Within this context, there has been an increased interest in determining the appropriate BC to study the flow dynamics of

Newtonian liquids. An increasing number of research groups resulted in a large number of experimental, computational, and theoretical studies about the slip BC. From these studies, it is inferred that liquid molecules can slip and have a non-zero velocity at a solid surface[16,17]. Reported results demonstrate slip length span a wide range of values from Angstroms to micrometers which cover molecular diameter scale to ultra-hydrophobic surfaces[18,19]. Substantial slip develops in non-wetting situations with slip lengths of 10–50 molecular diameters on smooth hydrophobic surfaces[20]. In general, boundary slips are described with a slip length $b$, interfacial slip velocity, $V_s$, liquid viscosity, $\eta$, and tangential viscous stress, $\sigma_{zx}$, at the solid wall[21]. Experiments and computer simulations revealed that the value of liquid slip (slip length) strongly depends on chemistry and morphology of solid surface which can be described by two main factors: (1) the interaction between the liquid and solid surface (2) and the roughness of the solid surface[22,23]. However, precise measurement of dynamics slip length remains a big challenge and no combined theory/experiments have so

[1]Nanotechnology and Integrated Bioengineering Centre (NIBEC), School of Engineering, Ulster University, Jordanstown, Shore Road, Northern Ireland BT37 0QB, UK. [2]Healthcare Technology Hub, Ulster University, Jordanstown, Shore Road, Northern Ireland BT37 0QB, UK. [3]Department of Polymer Science and Engineering, Chonnam National University, Gwangju 61186, Republic of Korea. ✉e-mail: a.farokh-payam@ulster.ac.uk; dlee@chonnam.ac.kr; n.bhalla@ulster.ac.uk

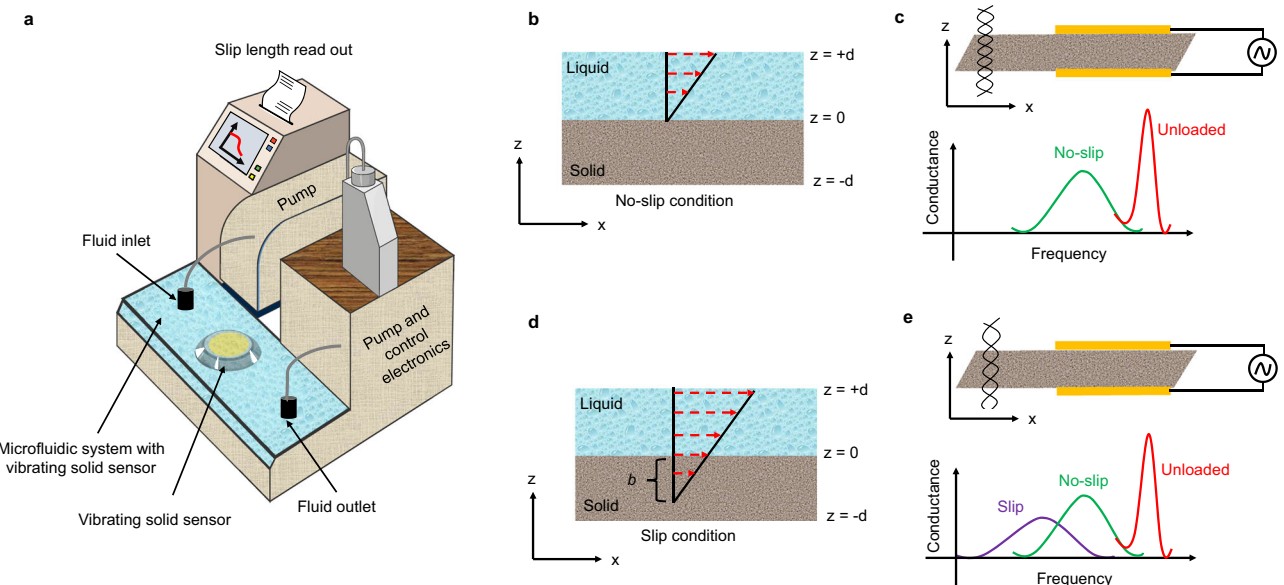

**Fig. 1 | Schematic of boundary conditions and experimental setup.**
**a** experimental setup consisting of vibrating solid, microfluidic system, readout and control electronics; **b** no-slip con- dition for the fluid flow, **c** state of sensor and its response in no-slip condition; **d** slip condition showing the slip length 'b' and **e** state of sensor and its response in slip condition.

far investigated the interrelationship between the vibration and wettability of the solid surfaces, flow rate and the interface slippage.

One of the difficulties in experimental measurement of a boundary slip at interface is due to the need of a very sensitive apparatus with respect to the shear stress[21]. Molecular dynamics (MD) provides a complement tool for estimating a boundary slip with capability of isolating effects of surface chemistry from the effects of surface roughness[21,24–26]. However, majority of these methods are limited to a non-vibrating system and cannot be used for dynamic situations (i.e., surface vibrating systems) of static or dynamic situations[11,27]. In principle, the quartz oscillates in response to an AC current, and the friction which occurs at the solid surface is related to the acoustic shear-wave[28]. Such a system forms a mass sensitive harmonic oscillator resulting in a change of resonant frequency of oscillation, $f$. In addition, when a fluid is in contact with a solid surface, the transverse waves propagate into the fluid, which results in energy dissipation depending on density and viscosity of the fluid[29]. The energy dissipation is dependent on the interfacial films or molecules deposed on the solid surface, and gives a useful quantitative parameter, i.e., damping rate, $D$. For a hydrodynamic system, most of the energy is either dissipated in the liquid environment or in the contact zone between the liquid and solid surface, while former has more contribution[27]. In another work, it was reported that the energy dissipation is not only determined by the adsorbate, but also by the link between binding molecules and solid surface[30,31]. In this regard, the dissipation factors can be used to understand the relationship among the adsorbate, hydrodynamics, and boundary slip. Historically, in QCM analysis, dissipation and hydrodynamics effects have been often disregarded although some studies show they affect the output of QCM measurements[32].

Given that the resonant frequency shift and energy dissipation are mutually related, and differ by the adsorbate, binding linkage, and hydrodynamics, it is very obvious to study more concrete theoretical background for boundary slip models by analyzing experimental data obtained by QCM under hydrodynamic situations. Within this context, we dedicate our particular attention to the effect of surface wettability and binding adsorbates to a solid surface on the boundary slip behaviors. We first investigate non-vibrating and vibrating systems, and extend the models to hydrodynamic situations where fluids move with certain velocities, followed by fluid inertia which involves the change of the resonant frequency shift and energy dissipation within the microfluidic QCM devices. To argue for this statement, we focus on calculating the dynamic slip length with a newly proposed theoretical model by varying surface hydrophobicity and binding adsorbates on the solid surface.

## Results

To elucidate the gliding of liquid on the vibrating solid, we used water as a liquid and quartz crystal as a piezoelectric material (vibrating solid) to study the slippage. The quartz was coated with gold on both sides for electrical contacts. Figure 1 shows the scheme for boundary conditions and the experimental setup. Essentially, Fig. 1a shows a microfluidic system integrated with a pump and a vibrating solid surface (quartz crystal). Figure 1b, c demonstrates the no-slip condition and state of QCM in no-slip condition, while Fig. 1d, e shows slip condition and the state of the QCM.

The surface of the quartz was functionalized by MPTS, plasma treatment, Octyl silane, and MPTS-Octyl silane as shown in Fig. 2a. The functionalization by MPTS− Octyl silane, MPTS, and Octyl silane treatments on the quartz crystals was confirmed by Fourier transform infrared spectroscopy (FTIR). Note that all FTIR spectra are shared in supplementary information. The FTIR spectra of MPTS show a characteristic absorption band at 2934 cm⁻¹, corresponding to Si−OCH₃ group. The sharp peak at 1065 cm⁻¹ and the broad peak at 3342 cm⁻¹ clearly indicate Si−O−C and O−H stretching. In case of the Octyl silane treatment, the CH₂−CH₂ binding peak and the stretching mode of Si−C bond are observed around 765 cm⁻¹ and 960 cm⁻¹, respectively. These main characteristic peaks are also observed in the surfaces coated with MPTS-Octyl silane. From these results, we confirmed that the MPTS and Octyl silane functionalization on of the vibrating quartz surface was successfully achieved. The static contact angle of the water drop on these surfaces differ depending on the molecules coated on the surface (Fig. 2b). The contact angles on the MPTS-coated and plasma treated sensor surfaces were measured to be <45°, confirming that the hydrophilic treatment was successfully achieved. Both the substrates coated with Octyl silane and MPTS-Octyl silane showed around 95°, indicating that the hydrophobic layer was created on the surface. These wetting conditions are with respect to untreated QCM substrate whose contact angle was found to be around

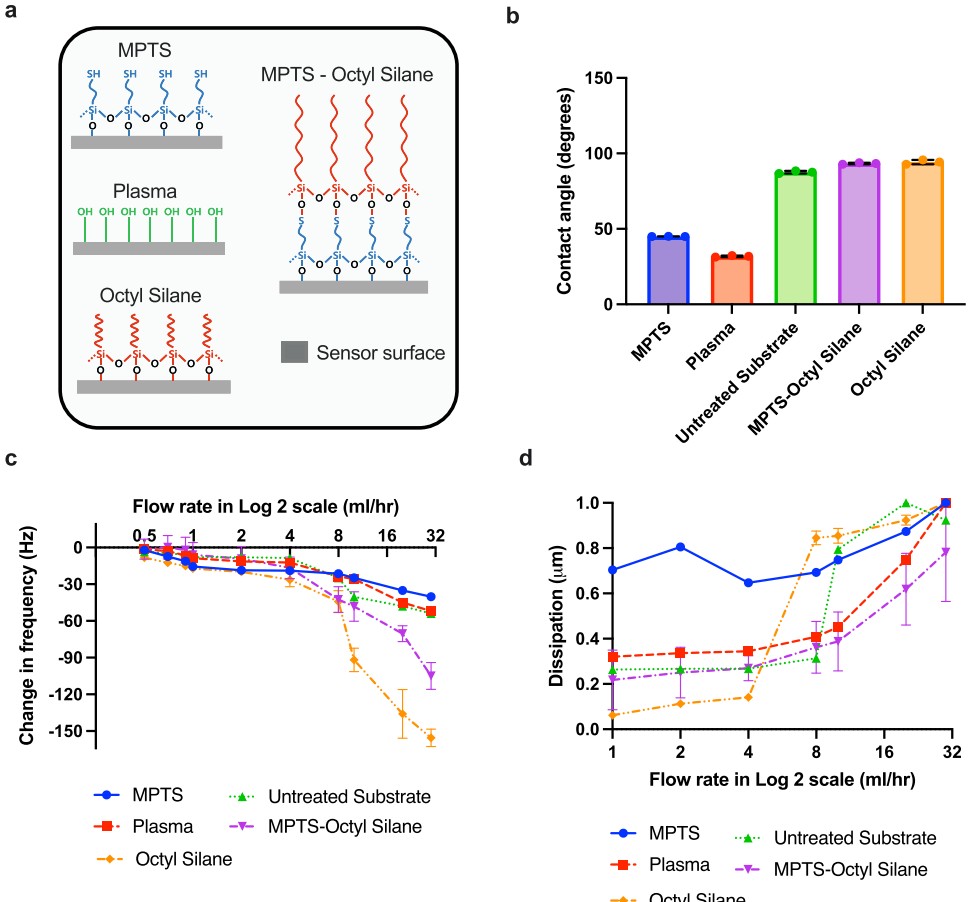

**Fig. 2 | Vibrating surface state and sensor response. a** Various types of modifications on the surface of the sensor creating either hydrophobic (MPTS-Octyl Silane and Octyl Silane) or hydrophilic surface (MPTS and Plasma) compared to untreated surface. It should be noted that the Fig. 2a is schematic representation and not an image showing chemical structures on the sensor surface. **b** Contact angle measurement from treated (MPTS-Octyl Silane, Octyl Silane, MPTS and Plasma) and untreated sensor surfaces. **c, d** The changes in frequency and dissipation of the sensor surface from various treated and untreated surfaces. Note that the error bars in **b**–**d** show the standard deviations.

87°. The difference in contact angle depending on chemicals confirms the presence of hydrophilic and hydrophobic groups on the surface. Surfaces with different wettability were developed (with varying degrees of hydrophilicity and hydrophobicity) and confirmed with contact measurements as the relationship between contact angle and liquid slip is well-known in literature[15,25]. Essentially, the surfaces with higher contact angle have more slippage compared to surfaces with lower contact angle.

The resonant frequency shift and energy dissipation depending on different molecules deposited are shown in Fig. 2c, d. We analyzed the data while increasing the flow rate of water in the range of 0.5 ml to 30 ml/h. In all cases, the shift of resonant frequency gradually increases with increase in the flow rate (Fig. 2c). It is worthy to note that the amount of decrease in the resonant frequency shift becomes more obvious on more hydrophobic sensor surfaces. It was reported that the hydrophobization of surfaces decreased the acoustic coupling of the surfaces to the liquid, resulting in higher frequency change and lower energy dissipation (or in other words, higher bandwidth) by reduced drag forces on the surface[33]. In our case, we could observe significantly reduced dissipation at low flow rate ranges for hydrophobic surfaces such as Octyl Silane and MPTS-Octyl Silane compared to the hydrophilic surfaces coated with MPTS and Plasma (Fig. 2d). Especially, the MPTS coated surface showed the highest energy dissipation at low flow rates owing to the strong attraction between the water and the hydrophilic surface. When the flow rate increases, the frequency change and energy dissipation generally increase owing to the fluid

inertia. The fluid inertia is highly related to the attraction between the fluid and the surface, thereby, the liquid slip on the surface can significantly affect those resonance parameters. Therefore, slip is expected to occur at the interface and the amount of slip depends on the interaction between the liquid and solid surface. In essence, liquid slip has consequences on both the frequency shift and energy dissipation, especially when the slip length is comparable to the inertial length of the liquid[21]. At a static condition of liquid system on sensor surface, the frequency shift and the energy damping decrease with the slip length[11]. In such conditions of a fluid system, fluid inertia is negligible due to the fact that the fluid velocity is zero. However, in our case, fluid inertia is also responsible for the resonant frequency of oscillation and energy damping so that they tend to increase with increasing flow rates. Therefore, frequency and dissipation changes associated with the change in the flow rates are attributed to the changes in liquid slip at the interface of the vibrating solid and water. We model this slip behavior using small approximation load in Eq. (1) to express frequency and damping ratio in terms of the load impedance of the vibrating solid.

$$Z_L = \frac{-i\pi Z_Q \left(\triangle f + \frac{if\triangle D}{2}\right)}{f_0} \qquad (1)$$

where $\Delta D$ is damping rate, $f_0$ is the natural resonant frequency of the vibrating solid, $f$ is the frequency of the vibrating solid when subjected to fluid, $\Delta f$ is frequency shift $f$-$f_0$, $Z_Q$ is the complex impedance of the

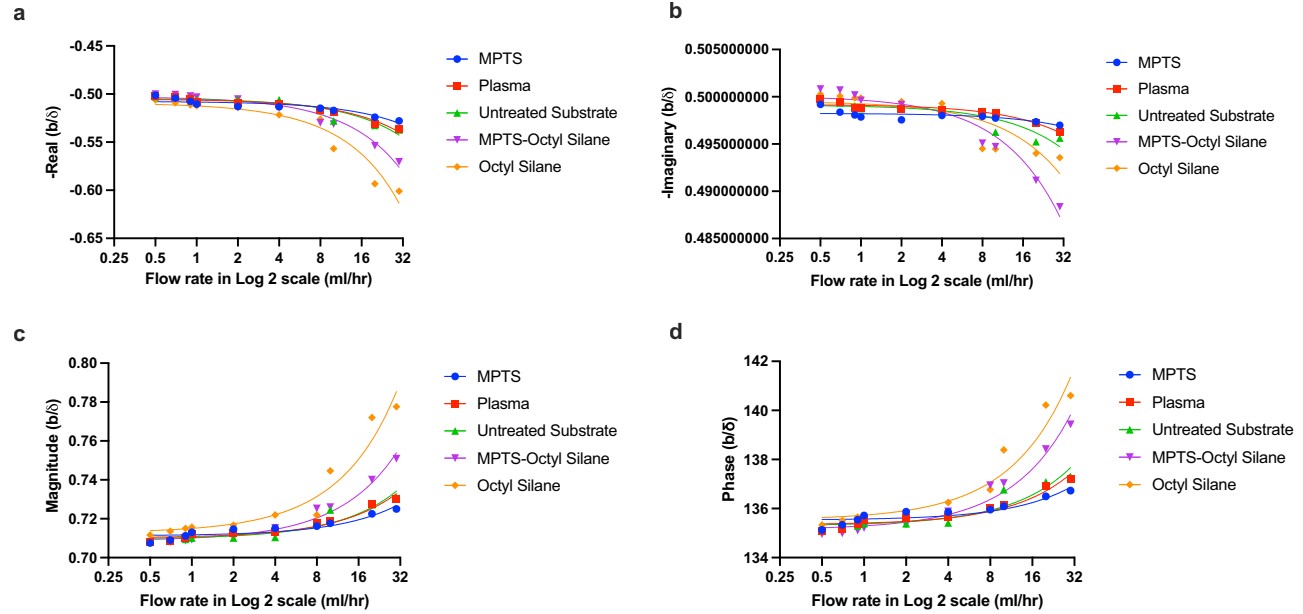

**Fig. 3 | Liquid slip modeling. a** The real part, **b** imaginary part, **c** magnitude, and **d** phase, in radians, of the liquid slip with change in flow rates of treated (MPTS-Octyl Silane, Octyl Silane, MPTS and Plasma) and untreated sensor surface. Note:

We have considered **a** negative real and **b** negative imaginary slips to plot as we have considered the surface of the vibrating solid as (0,0) plane in the x-y coordinate system.

vibrating material (in our case, quartz). To elucidate the slip behavior we have considered three cases; (1) vibrating solid-liquid interface boundary condition without slip, (2) vibrating solid-liquid interface boundary condition with slip, and (3) vibrating solid-liquid interface boundary condition with slip including inertia force. The inertial force which is considered here is due to the contribution of inertia of the first water layer to the momentum/energy transfer at the vibrating solid/liquid interface.

The vibrating solid-liquid boundary conditions considered for the aforementioned cases include the reference to a fluid system with stokes flow regime in contact with a vibrating plane wall. The impedance described in Eq. (1) relates the shear stress on the vibrating solid surface due to variation in the velocity of the liquid in contact with the surface. In addition, perturbation expansion can be applied to the frequency change of the unloaded vibrating solid. We also consider that the fluid with density $\rho_f$ and shear viscosity $\eta_f$ obeys the Navier–Stokes equation for a Newtonian liquid while assuming continuity and incompressibility of the vibrating solid-liquid interface system. Equations describing detailed fluid velocity and solutions to it with boundary conditions which allow us to observe the displacement of vibrating solid substrate are described in details within supplementary information. Considering the above boundary conditions, the case 1 depicts a no-slip condition where the fluid velocity and the velocity of the vibrating solid should be equal at the boundary solid-liquid interface, equivalently, and their displacements can be matched providing us with a load no-slip impedance ($Z_L^{ns}$) expression (Eq. (2)).

$$Z_L^{ns} = (1-i)\sqrt{\frac{\rho_f \eta_f \omega}{2}} \qquad (2)$$

where $\omega$ is the frequency of vibrating solid. For the case 2, slip length (*b*) is added to the no slip condition by applying Ellis-Hayward method and Taylor expansion (more details are in the supplementary information). This yields a new empirical expression (Eq. (3)) for the liquid slip. It should be noted that within this equation, the slip is normalized by characteristic decay length of the fluid, called as penetration length ($\delta$). This is primarily done to define the liquid slip as a dimensionless number, which generally helps in reducing the

variables that describe a system by facilitating better correlation of a system to the physical phenomenon.

$$\frac{b}{\delta} = (1+i)\frac{\left(Z_L^s - Z_L^{ns}\right)}{2Z_L^{ns}} \qquad (3)$$

In our experiment, we have directly measured the changes in the frequency and damping rate (dissipation) of the vibrating solid. Therefore, by using Eq. (1) to calculate $Z_L^s$ and from Eq. (2) we calculate $Z_L^{ns}$. The ratio between a slip length and penetration length can be obtained from Eq. 3 and consequently a slip length can be calculated. For the case 3, we consider the friction force per unit area exerted by the liquid on the vibrating solid which is the sum total of shear stress exerted by the liquid at the solid-liquid interface and the inertia of the solid-liquid interfacial liquid. This introduces a new parameter $l_a$, defined as normalized inertia length of the solid-liquid interface, within the Eq. 3 which changes the dimensionless number $b/\delta$ presented in Eq. (3) to Eq. (4). Essentially, the inertia length is defined as ratio of surface number molecular density of first liquid layer to the density of molecules in the bulk liquid. Detailed derivation of Eq. 3 as well as validation and comparison between our proposed equations with well-established and recently proposed equations[21,34] for slip length are shared in the supplementary information.

$$\frac{b}{\delta} = (1+i)\frac{\left(Z_L^s - Z_L^{ns} - i\omega\rho_f l_a\right)}{2\left(Z_L^{ns} + i\omega\rho_f l_a\right)} \qquad (4)$$

In Fig. 3 we plot a relation between our flow rates and the normalized slips (Eq. (3)). Specifically, we show the real part (Fig. 3a), imaginary part (Fig. 3b), magnitude (Fig. 3c), and phase (Fig. 3d) of the dimensionless numbers which represent the liquid slip.

Note that we have considered negative real and negative imaginary parts to plot in the Fig. 3a, b, with respect to the surface of the vibrating solid which is considered as (0,0) plane in the x-y coordinate system (Fig. 1d). Therefore, the negative real and imaginary values refer in the figures to the physical system below the surface of the vibrating solid. It is clear from the trend that as flow rate increases, the

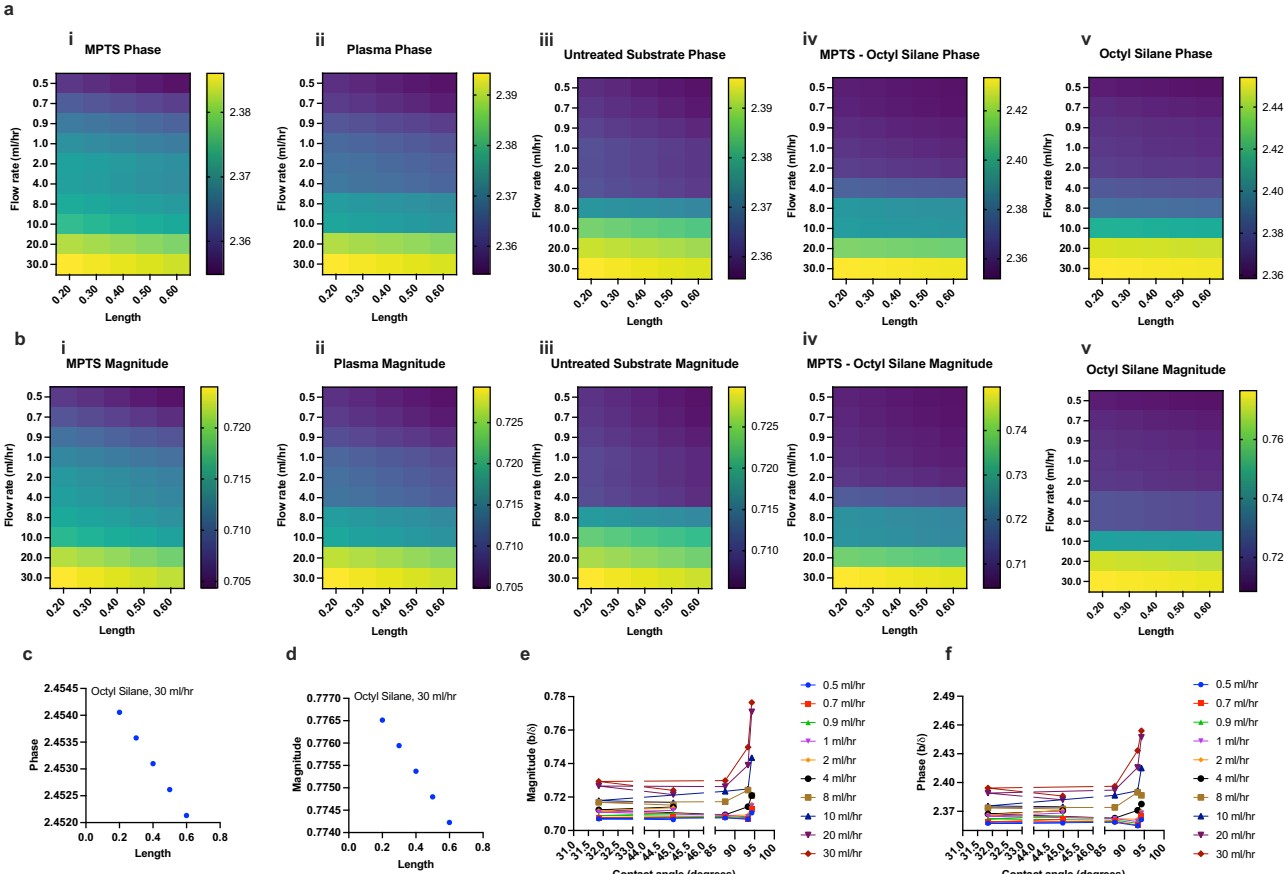

**Fig. 4 | Flow rate vs liquid slip. a** A heat map with change in the phase of the slip for different surface conditions (i. MPTS, ii. Plasma treated, iii. Untreated substrate, iv. MPTS-Octyl silane, v. Octyl silane); **b** a heat map with change in the magnitude of the slip for different surface conditions (i. MPTS, ii. Plasma treated, iii Untreated substrate, iv. MPTS-Octyl silane, v. Octyl silane); **c** inertial length vs. phase changes in the slip for Octyl silane treated surface; **d** inertial length vs. changes in the magnitude of the slip for Octyl silane treated surface; **e** Changes in the magnitude of the slip upon change in the contact angle, and **f** depicts changes in the phase of the slip with change in the contact angle. Note that phase in **f** is presented in the units of degrees and the units of length in **a**, **b** is nm.

liquid slip also increases including the rate of slip which increases at higher flow rates. This is because at higher flow rates above 8 ml/h (Reynolds number, $Re$ = 1.11), the flow becomes inertia dominant as the $Re$ is essentially the ratio of inertial to viscous forces exerted by the liquid. This also supports our observation that dissipation sharply increases upon 8 ml/h ($Re$ = 1.11).

To further analyze the effect of the inertial forces, we vary the inertial length ($l_a$) of the vibration solid-liquid interfacial system as described in the Eq. (4) from 0.2 to 0.6 nm and then plot the changes in the slip against the flow rate. The phase and magnitude of the normalized slip ($b/\delta$) is plotted in the Fig. 4 for all surface conditions of the vibrating solid. The phase plots, see Fig. 4a, clearly indicate change in the rate of the slip above $Re$ = 1.11 (flow rate of 8 ml/h) where inertial forces are more dominant than the viscous forces exerted by the liquid. The magnitude of the slip, compared in the Fig. 4b, depicts that as the surface becomes hydrophobic, there is more slip with increase in the flow rate as less interaction with surface leads to the increase of liquid molecules velocity at surface. However, we cannot completely resolve the effect of flow on the slip at flow rate below 4 ml/h. This is due to minute limitation associated with the resolution of the electrical measurement system. For instance the frequency and dissipation changes of the vibrating solid measured are around 20 Hz for flow rates between 0.5 and 4 ml/h which are small changes for an electrical system. While the inherent noise of the system is less than 0.1 Hz and the experiments are conducted in electro-mechanical noise free background, interference from combination of liquid, applied voltage and quartz, typically up to 0.5–1 Hz, are difficult to avoid. In addition,

treatment of plasma on the surface of the solid is short-lived, i.e., the formation of −OH groups on the surface is less stable as compared to the MPTS treatment. Therefore, the vibration changes of 2−3 Hz in flow rates less than 4 ml/h are non-trivial to distinguish for isolating the explicit contribution of their respective slips. For the aforementioned reason, short lived −OH groups, the slip in the plasma surface is also a bit higher than the surface coated by MPTS while its measured static contact angle is lower. In Fig. 4c, d, we also show the inertial length changes with the phase and magnitude of the slip. As an example of octyl silane plotted in these sub figures, we see a linear decrease in the liquid slip with increase in the inertial length. Note that here the inertial length refers to the length of first water layer on the solid surface which contributes to the momentum/energy transfer at the liquid/solid interface. Hence, it can be inferred that increasing the length of first nano/angstrom scale layer of water molecules at the surface leads to a decrease in the molecular velocity of fluid interacting with this layer. We also find the relationship between the static contact angle and the magnitude and phase of the slip as shown in Fig. 4e, f. We observe that as static contact increases, the slip also increases which means there is a direct relation between hydrophobicity of the surface and slip. This is due to the fact that at hydrophobic surfaces water molecules are expelled from the vicinity of the surface and replaced by several molecular-size hydrophobic layers[35]. This can lead to the increase of the flow velocity at the vicinity of solid surface. In addition, we can also see that with the increase in the flow rate the values of slip increases. Note that the slip for MPTS is less as compared to oxygen plasma whose contact angle is lower than MPTS. This is due to the fact that

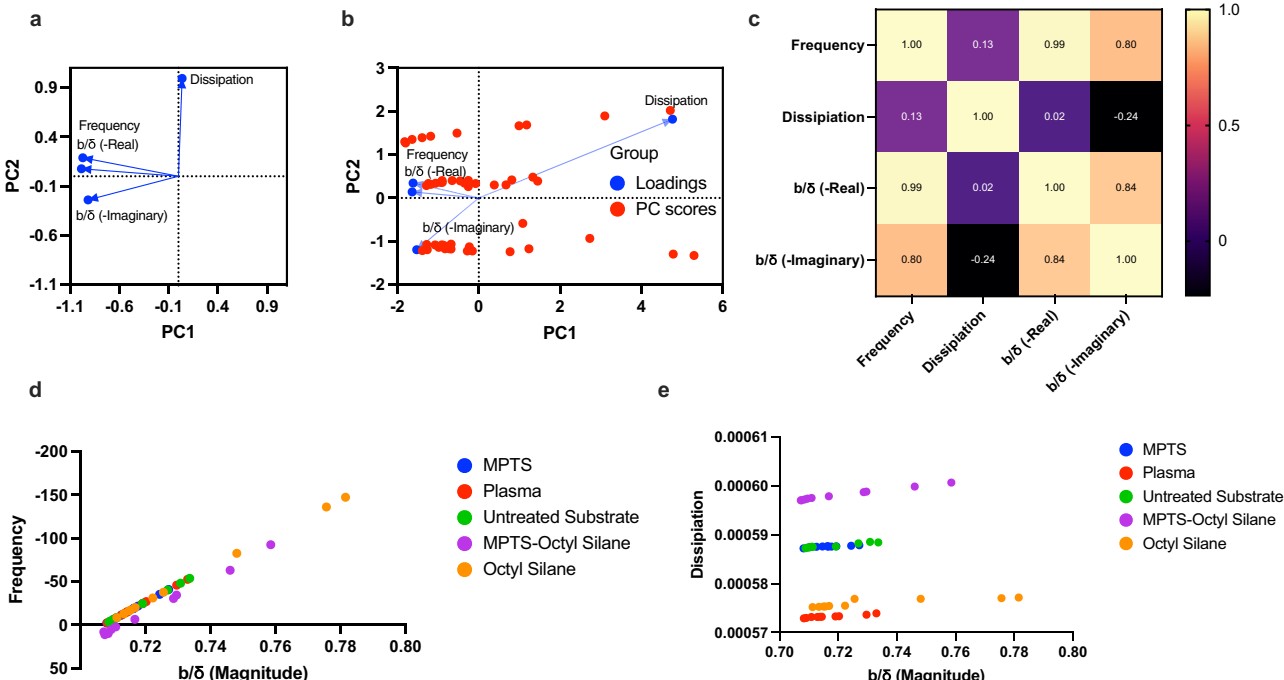

**Fig. 5 | Statistical evaluation of theory and experiment. a** Loading plot generated by principal component analysis showing relationship between different variables which influence the sensor response. **b** Loading plot with principal component scores of the variables allowing visualization of relationships between liquid slip and sensor response. **c** Correlation matrix depicting relationship between liquid slip and sensor response. **d** Frequency vs slip for all dataset measured in experiments and analyzed by the proposed model and **e** dissipation vs slip for all dataset measured in experiments and analyzed by the proposed model.

oxygen plasma treatment is short lived (as also mentioned earlier) and during the flow of the fluid (during experiment) some of the –OH bonds may not be present on the surface for longer duration and as a result it becomes slightly more hydrophobic than what its contact angle suggests. Nevertheless, there are significant number of –OH bonds to consider it hydrophilic compared to untreated surface as suggested by slip values, contact angle, FTIR and AFM measurements.

To further visualize the relationships between the experimental measurements (frequency, dissipation, contact angle) and the slip deduced by our analytical model, we performed principal component analysis (PCA) and multiple variable analysis (MVA), see Fig. 5. This is because PCA allows us to understand and find the important patterns between analyzed variables. The two principal components (PC1 and PC2) of the PCA are selected using horn's parallel analysis. Figure 5a shows the loading plot generated from PCA analysis. The plot shows that frequency and the real part of the slip are closely associated with while the imaginary part of the slip contributes less towards changes in the slip. A more clear relationship is demonstrated by Fig. 5b where we plot the PC scores or data points which allow for the projection of the original data into the two-dimensional space defined by two of the selected PCs. Usually, this is a visual representation of the dimensionality reduction goal of PCA where points close to each are considered to have a direct correlation with others. However, we can also associate relationships between each group using such plots. For instance, as aforementioned, we can affirm that frequency change in the sensor response and real part of the slip are directly proportional to each other. Similarly, since the dissipation and imaginary part of the slip lie on the same line but at opposite ends, we can qualitatively confirm that dissipation measured by the sensor is inversely proportional to the imaginary part of the slip. We also perform MVA to further support our observation of relationships between the frequency, dissipation, real and imaginary part of the slip. The MVA, see Fig. 5c, revealed correlation coefficients of 0.989 (between real part of the slip and the frequency), 0.799 (between imaginary part of the slip and the frequency) and −0.235 (between imaginary part of the slip and the

dissipation). The PCA and MVA thus support the comprehensiveness of our developed model, where real part of the Eq. (4) mainly influences the frequency of the vibrating solid while the imaginary part contains more signatures of dissipation energy on the surface of vibrating solid. To verify our analysis and hyphothesis, we plot the measured frequency shifts and dissipation versus slip. From Fig. 5d it is clearly obvious that independent of solid properties, there is a linear trend between frequency shift and slip. However, it is worthy to mention that the same slip is obtained at different flow rates for different materials (depending on the hydrophilicity/hydrophobicity of solid). On the other hand, from Fig. 5e, it can be deduced that in addition to the relation between dissipation with slip, there are other factors such as solid properties (i.e. hydrophilicity/hydrophobicity), molecular binding and adsorbate which affect the dissipation in the fluidic systems.

We demonstrate a new model for the characterization of liquid slip behavior on vibrating solid surfaces. We show that the slip increases with increase in the flow rate and that leads to changes in the resonant frequency of the vibrating solid and dissipation of energy on its surface. In addition, the developed equations can also be reliably used to compute slip in static mode, i.e., when the liquid is not flowing, see validation and comparison of our model with other models in literature used to calculate liquid slip in static conditions in supplementary information. Our work has potential to revolutionize the current biosensing measurements where most often the sensor response is associated with a single stimulus. For example, in the case of quartz crystal microbalance, the frequency shifts are often exclusively associated with mass binding on the sensor surface. The effect of such this has at times led researchers to associate faster binding of the biomolecules after observation of higher change in frequency upon change in flow rate. For instance, there are works which did not study what happens on the quartz crystal microbalance surface without any biomolecule, whilst associating the frequency changes upon flow rate with association/dissociation of the biomolecule[36]. In contrast, our work shows changes in frequency of the quartz crystal microbalance

simply due to change in flow rate of the buffer which is associated with liquid slip and not exclusively faster biomolecular binding. Similar concepts can also be extended to isolate slip effects in other flow based sensing systems such as surface plasmon resonance systems or routine electrochemical sensing platforms integrated with automatic fluidic systems where effects of flows may as well be associated with sensor sensitivity and binding affinities[37–39]. In addition, the equations developed also serve as a fundamental basis for the detection of liquid slip length using vibrating solid based sensors.

## Methods

### Materials used
AT-cut 10 MHz gold-coated quartz crystals (QSX301) with polished surfaces for one-sided contacting were purchased from Open QCM, Italy. All chemicals were of analytical grade and were used as received. All aqueous solutions were made with double deionised water. (3-Mercaptopropyl-trimethoxysilane (MPTS) (175617, Sigma-Aldrich, Korea) and Trichloro(octyl) silane (235725, Sigma-Aldrich, Korea) were used to impose hydrophilicity and hydrophobicity on the quartz crystals, respectively. In addition, some parts of Fig. 2a were adopted from smart.servier.com.

### Fuctionalization of the sensor surface
Prior to use, the quartz crystals were cleaned by exposure to a solution of 5:1:1 of ultra pure water, hydrogen peroxide and ammonia preheated at 75 °C for 10 min. Cleaned quartz crystals were rinsed with DI water, ethanol and DI water, and dried with nitrogen gas. Plasma treatment was carried out to create hydrophilic hydroxyl groups on the quartz crystals using a plasma reactor, purchased from Harrick Plasma, USA, in the air environment. For a thin-layered hydrophilic coating with MPTS, 1 vol% of MPTS was added to ethanol to prepare MPTS solution. The quartz crystals were immersed into the solution and incubated overnight at room temperature. They were rinsed thoroughly with DI water and dried with nitro- gen gas. A hydrophobic self-assembled monolayer (SAM) of Trichloro(octyl)silane was deposed on the quartz crystals using a chemical vapor deposition (CVD) technique. The cleansed quartz crystals were placed in a vacuum oven at 100 °C for 30 min with a glass dish containing a few drops of Trichloro(octyl) silane. For creating a double layered hydrophilic-hydrophobic coating, Trichloro(octyl) silane was deposed using a CVD method on the MPTS coated quartz crystals.

### Surface characterization
The chemical composition of the surface functionalization was evaluated by Fourier transform infrared spectroscopy (FT-IR, JP/FT/IR-4200, Jasco, Japan) to confirm the deposition of hydrophilic and hydrophobic chemicals. The contact angle on the surface- functionalized quartz crystals was measured by contact angle goniometer (L2004A1. Ossila, UK). Water drops are gently placed on the functionalized substrate to measure the contact angle. The quartz crystals were mounted in titanium chambers and used with an QCM-D sensor system (Open QCM, Italy). The chamber temperature was an maintained at 25 °C. DI water was used to flow it into the chamber by a syringe pump (Fusion 200-X, Chemyx, USA). The flow rates were increased from 0.5 ml to 30 ml/h. The atomic force microscopy (data in supplementary information) was performed using Park systems XE-100 in non-contract mode. The size of the images acquired was 50 μm × 50 μm at a scan rate of 0.3 Hz.

### Data analysis
Heat maps are plotted using a graphpad prism software. The Principal component analysis (PCA) is performed by selection the principal components using horn's parallel analysis. This method principal accounts for variance in the data due to random error or noise. The first step in the analysis process is to perform PCA on the original experimental dataset and determine the eigenvalues for each of the PCs. Next we simulate a dataset with the same number of variables and observations as the original data using the in-built monte-carlo simulation tool in the graphpad prism software. Then we perform PCA on the simulated dataset and determine the simulated eigenvalues. We repeat the simulation/PCA processes a 1000 time to calculate eigenvalues for each simulation. The average and 95th percentile of the eigenvalues for each PC across all simulations are then compared with the actual eigenvalues (from the original dataset). Finally, we retain components with eigenvalues greater than the 95th percentile of the eigenvalues from the simulations. Note that the PC1 and PC2 comprises 89.41% of cumulative proportion of variance. All other multiple variable analysis such as correlation matrix and loading points plotted from PCA are performed using the in-built functions in the graphpad prism software.

## Data availability
Original raw data files related to AFM were generated at large scale research facility of School of Polymer Science and Engineering, Chonnam National University. These are available from all corresponding authors upon reasonable request. However, we provide the source data for all figures in the paper and supporting information, where applicable. Essentially, source data for Figs. 2b, 2c, 2d, 3a, 3b, 3c, 3d, 4a, 4b, 4c, 4d, 4e, 4f, 5a, 5b, 5c, 5d, 5e, S4, S5, S6a, S6b, S7a, S7b, S8a, S8b, S9, S10, S11a, S11b, S11c and S12 are provided with the paper. Source data are provided with this paper.

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

## Acknowledgements
The authors would like to thank support from National Research Foundation of Korea (NRF) grant funded by the Korea government (MSIT) (NRF-2021R1C1C1014042) and the Department for Economy, Northern Ireland through US-Ireland R&D patnership grant No. USI 186.

## Author contributions
N.B. did initial experiments and conceived the concept and experiment design with D.L.; A.F.P. developed analytical model, theory and implemented them on experimental data for the liquid slip; B.K. performed experiments under close supervision of D.L.; A.F.P., D.L., and N.B. prepared the first draft and equally contributed towards data analysis and methodology development; N.B. prepared all figures; All authors also equally contributed in editing the manuscript during revisions.

## Competing interests
The authors declare no competing interests.
