## [Peer Review File · Nature Communications]

Unraveling the liquid gliding on vibrating solid liquid interfaces
with dynamic nanoslip enactmentREVIEWER COMMENTS

Reviewer #1 (Remarks to the Author):

Payam et al develop an analytical model to quantify the liquid slippage at the interface with vibrating solids. The main novelty comes from the ability of the model to quantify the impact on inertial forces -an imposed liquid flow- on the slippage something not done before. Additionally, the method could offer a more direct and easy route to quantifying slippage at interface, something notoriously difficult.

My biggest concern with the proposed method is that it seems to lack independent validation. As the paper stand, the model developed is used to calculate slippage and dissipation from experimental data, but there is no way of independently verifying the results, let alone its precision. The model seems physical, makes sense and may well provide interesting novel insights, but without independent benchmarking or verification it is just that, a possible model. Have the authors considered validating with surfaces and static conditions where slippage is known?

There are also some serious experimental issues that should be addressed: since the surfaces are functionalised, do they age or are the results reproducible on a given surface over multiple flow cycles? Do all the surface have a similar roughness/functionalisation quality? Does it make sense to compare them directly?

Finally, the paper as currently written is often unclear with information presented in a convoluted and confusing manner. This is not a criticism of the information itself, but rather of the manner it is presented. For example:

-The correlation between CA and slippage, well know in the specialist literature, is never properly introduced making the point of comparing different functionalisation not obvious.

-Methodologies are often introduced before explaining why they are needed (e.g. PC analysis)

-Fig 1c offers a nice schematics of the QCM-D frame, but this is unhelpful. The reader is left wondering about the actual sensor. What are its more of vibration? What are the in-plane and out of plane motions and their magnitude? What is the relative velocities of vibration in comparison to the imposed steady flow? How does the setup compare to the situation in (a) and (b)?

-Fig 2: (b-d) could be moved to SI since their are merely a control. However, e is confusing (why not a bar diagram) and the color coding in (f-g) makes it impossible to correlate CA with the observations.

-Fig 4 e suggests no particular correlation between CA and slip magnitude, but rather only a clear signature of flow rate. This deserves more comments discussion since it is not expected.

Reviewer #2 (Remarks to the Author):

Slip at the solid-liquid interface is an interesting topic in many research areas. This work presents a noval experiment approach which incorporates the QCM (quartz crystal microbalance) sensor into a micro-channel of fluids. Meanwhile, a relationship between

the resonant frequency shift and energy dissipation ratio of the sensor with the slip length is established. The predictions of the model are validated by measurements by using functionalized sensors with different contact angles in a wide range of flow rate. The main findings of the study is the contribution of an inertial component to the slip length at higher flow rates.

There are a few points need to be clarified in revision before it is accepted for publication.

1. Did you measure the pressure difference between the inlet and outlet in the microfluidic experiments? Based on fluid mechanics of Newtonian fluids, the effective slip length can be calculated from the test results of flow rate and pressure drop. The slip length estimated with the QCM measurement data and Eq. (4) should be compared with the effective slip length.

2. The dissipated energy in the microfluidic transforms into heat and makes the temperature of fluid rise. Did you account the temperature effect in derivation of Eqs.(3) and (4)?

3. As shown in Fig.3, the slip length increases with flow rate monotonically. If flow rate is much larger than 32ml/hr, how the slip length changes?

4. The authors assume that slip occurs between the first liquid layer and the solid surface. Is this assumption always true? For water, there are experiment measurements showing that the water molecule nearest to the solid surface are solid-like.

5.References [3] and [15] missed information.

Reviewer #1 (Remarks to the Author):

Payam et al develop an analytical model to quantify the liquid slippage at the interface with vibrating solids. The main novelty comes from the ability of the model to quantify the impact on inertial forces -an imposed liquid flow- on the slippage something not done before. Additionally, the method could offer a more direct and easy route to quantifying slippage at interface, something notoriously difficult.

We thank the reviewer for appreciating our work and for providing useful suggestions. In response, we have implemented all suggestions from the reviewer. Below is our point-wise guide describing the changes we have made in the revised manuscript.

My biggest concern with the proposed method is that it seems to lack independent validation. As the paper stand, the model developed is used to calculate slippage and dissipation from experimental data, but there is no way of independently verifying the results, let alone its precision. The model seems physical, makes sense and may well provide interesting novel insights, but without independent benchmarking or verification it is just that, a possible model. Have the authors considered validating with surfaces and static conditions where slippage is known?

*We thank the reviewer for this comment. We have now verified our developed model and experiments with 3 different models proposed in the literature (called McHale, Heyward and Ellis in our paper, referenced accordingly). Note that all these models have measured slip in static mode using QCM surfaces and our model is the first one which measure slip in dynamic conditions with insights into inertial length of the system. In addition, none of the systems in the literature use surface modifications similar to our functionalization. To validate our proposed equations, we have compared our equations with models in literature for the case of water resting on the QCM surface without flowing conditions, as well as ethanol and different concentration of glycerol for measurement of QCM response from different types of QCMs (with 5 MHz and 9 MHz as their natural frequencies). Note that the frequency of resonance in our work is 10 MHz. We have extracted the data from these QCMs works and used it to calculate slip using 3 models (where appropriate) and compared it with our method. Overall, we would like to summarize that we have compared different QCMs (other than what we used in our work), different solutions (ethanol and glycerol) and water which is used commonly for static slip characterization. This serves as an independent benchmark for testing our developed equations for work available in the literature. Below we have explained our detailed validation. **Please note we have added this validation in supplementary information.***

3 Models used for validation

1. [Model 1] **McHale**, Glen, and M. I. Newton. "Surface roughness and interfacial slip boundary condition for quartz crystal microbalances." *Journal of Applied Physics* 95.1 (2004): 373-380.

2. [Model 2] G.L. **Heyward** and M. Thompson. "A transverse shear model of a piezoelectric chemical sensor." *Journal of Applied Physics* 83.4 (1998): 2194-2201

3. [Model 3] S. **Ellis**, and G. L. Hayward, *Interfacial slip on a transverse-shear mode acoustic wave device*, *Journal of Applied Physics* 94.12 (2003): 7856-7867.

Reference in the proceeding text

4. Han Zhuang, Pin Lu, Siak Piang Lim, and Heow Pueh Lee, *Effects of Interface Slip and Viscoelasticity on the Dynamic Response of Droplet Quartz Crystal Microbalances*, *Analytical Chemistry* 2008, 80, 7347–7353.

1. Comparison with the work of McHale [model 1] where 10 MHz QCM was used.

Here, static slip of water on gold surface of the QCM is considered and we calculate the normalized slip length and compare the result with McHale [1]. Please note that we have used both dissipation and frequency shifts of the vibrating solid (QCM) to calculate the slip length (unlike McHale which does not consider dissipation effects). The figure below shows the comparison of slip lengths. Here in the figure a) we show the frequency shifts and dissipation measured by our new experiments. Figure b) compares the real value of the slip length in our model with the McHale's method.

Figure 1. a) Frequency and dissipation measured in our experiments b) comparison of slip length calculated from our experimental data with the slip length calculated by McHale.

We can clearly see that the slip for ethanol is higher than that for water, which is in agreement with the results in McHale's work [model 1] and Ellis [model 3]. Additionally, our calculated slip length has less than 3.7% difference for ethanol and less than 2.5% for water compared to McHale's work. Note that measurement using ethanol is non-trivial as ethanol is highly volatile and evaporates very quickly, which leads to more differences from measurement to measurement. Nevertheless, this shows that our method is well suited for slip measurements in static conditions (in addition to dynamic slip which we show in our work). The difference between our method and McHale can be attributed to the absence of inertial length and dissipation effects in their equation as well as limitation of McHale method to calculate only real value of slip.

2. Our equations tested with data generated by Zhuang et al. (reference 4 above) from 5 MHz QCM and compared with McHale's equation [model 1].

Here, we have extracted the frequency shifts from the work of Zhuang et al. (reference 4 above) for different concentrations of glycerol, see Figure 2a) below for average frequency shifts in their work. We have then used these frequency shifts to compare the McHale and our method by calculating the slip length. Interestingly, we see that McHale's work and our method have less than 2% difference for concentrations of glycerol above 90%. Below 90% concentration of glycerol, we had a maximum of 8.4% difference which suggests that as the solution becomes less viscous, dissipation and inertial effects are apparently easy to resolve using our developed equations. More studies are required to validate this relatively large difference (< 2% compared to < 8.4%) at lower concentrations of glycerol and here we just provide this extra information to elucidate differences.

Figure 2. a) Frequency and dissipation measured by Zhuang et al. (reference 4 above) b) comparison of slip length calculated from our experimental data with the slip length calculated by McHale using data in the work of Zhuang et al..

Irrespective of this, the comparison here again demonstrates that our model is versatile and it can utilize data from other works i.e. vibrating solids of different frequency and different solution (other than those tested in our experiments) and can be used to calculate slip length.

3. Our equations compared with work of Heyward [model 3] from 9 MHz QCM.

We have used the frequency shifts obtained by Heyward et al. for change in concentration of glycerol from 0-20% on a 9 MHz QCM. We then compared the changes in the magnitude of slip obtained by our equations and Heyward et al.. The maximum difference is less than 4.8% which is again due to consideration of dissipative and inertial effects in our equation. The trend of slip length calculated by Heyward's method and McHale method (shared above) remains the same % differences observed at low and high concentrations of glycerol tested in the irrespective papers.

Figure 3. a) Frequency and dissipation measured by Heyward et al. [model 2] b) comparison of slip length calculated from our experimental data with the slip length calculated by Heyward's method.

3. Direct comparison of slip length calculated by Ellis [model 3] using 9 MHz QCM

In Figure 4 below we also compare our method with Ellis's method where the authors have calculated the slip length for ethanol without consideration of dissipation effects.

Figure 4. Comparison of normalized slip length calculated with Ellis's method. Our method shows less difference in slip length (from one experiment to another) in comparison to Ellis's method.

All the above comparisons serve as independent validation of slip length calculated by our model. From the comparisons we can conclude not only our model is valid in different conditions including different QCM frequencies and solution concentration, but also due to consideration of the effect of dissipation and inertial length, our method is more accurate than the well-known established QCM equations to calculate the slip length in static conditions.

There are also some serious experimental issues that should be addressed: since the surfaces are functionalised, do they age or are the results reproducible on a given surface over multiple flow cycles? Do all the surface have a similar roughness/functionalisation quality? Does it make sense to compare them directly?

From the above comment, see below our actions specific to different questions.

Since the surfaces are functionalised, do they age or are the results reproducible on a given surface over multiple flow cycles?

We have performed the experiment at least 16 times on 3 different QCM substrates for each condition over the whole surface. Note that we have 5 conditions, where 3 of them are functionalized, 1 is bare substrate and the last one is simply the oxygen plasma treated substrate. We have mentioned that the oxygen plasma treated is short lived and as a result of which vibration changes of 2-3 Hz in flow rates less than 4 ml/hr are non-trivial to distinguish for isolating the explicit contribution of their respective slips in the plasma treated samples.

However, for MPTS, MPTS-Octyl Silane, Octyl Silane treatments on the QCM sensor surface we now access the effect of flow on the functionalized surfaces. We use FTIR before and after the exposure of functionalized QCM surface to multiple cycles (7 cycles) of flow rates ranging from 0 - 32 ml/hr. We observe that our functionalization remains intact and does not age during our experiment. Therefore, during our flow

measurement, all surfaces retain the functionalized groups. See above our FTIR results showing the surface condition before and after the aforementioned flow experiments. **We have added these figures in the supporting information** and mentioned in the main manuscript that the modified surfaces retain their functional group during the course of our flow experiments.

Do all the surface have a similar roughness/functionalisation quality? Does it make sense to compare them directly?

Yes, the functionalization quality is retained as demonstrated by FTIR. Regarding roughness we performed AFM measurement on 3 different functionalized surfaces suggesting that we can consider them as surfaces with different wettability. According to AFM roughness characterization, even after the use of sensor in flow, the QCM is in good state of roughness allowing us to compare the surfaces directly.

Finally, the paper as currently written is often unclear with information presented in a convoluted and confusing manner. This is not a criticism of the information itself, but rather of the manner it is presented. For example:

-The correlation between CA and slippage, well known in the specialist literature,

is never properly introduced making the point of comparing different functionalisation not obvious.

*We thank the reviewer for this suggestion and we completely agree that it is necessary to briefly introduce slippage and CA relationship in the manuscript, even though it is well known. **Therefore, we have added some discussion on CA in the introduction.** The added text is also shared below:*

The static contact angle of the water drop on surfaces differs depending on the molecules coated on the surface, see Figure 2(b). The contact angles on the MPTS-coated and plasma treated sensor surfaces were measured to be less than 45°, confirming that the hydrophilic treatment was successfully achieved. Both the substrates coated with Octyl silane and MPTS - Octyl silane showed around 95°, indicating that the hydrophobic layer was created on the surface. The difference in contact angle depending on chemicals confirms the presence of hydrophilic and hydrophobic groups on the surface. Surfaces with different wettability were developed (with varying degrees of hydrophilicity and hydrophobicity) and confirmed with contact measurements as the relationship between contact angle and liquid slip is well-known in literature. Essentially, the surfaces with higher contact angle have more slippage compared to surfaces with lower contact angle.

-Methodologies are often introduced before explaining why they are needed (e.g. PC analysis)

*We thank the reviewer for this comment. **We have now mentioned why PC analysis is required before explaining it deeply.** Before the start of the explanation we add following lines:*

To further visualize the relationships between the experimental measurements (frequency, dissipation, contact angle) and the slip deduced by our analytical model, we performed principal component analysis (PCA) and multiple variable analysis (MVA), see Figure 5. This is because PCA allows us to understand and find the important patterns between analyzed variables.

-Fig 1c offers a nice schematics of the QCM-D frame, but this is unhelpful. The reader is left wondering about the actual sensor. What are its more of vibration? What are the in-plane and out of plane motions and their magnitude? What is the relative velocities of vibration in comparison to the imposed steady flow? How does the setup compare to the situation in (a) and (b)?

*We thank the reviewer for this comment. The QCM is vibrating in its natural mode of frequency i.e. 10 MHz. With the help of schematic c and e now we explain the status of QCM under condition of slip and no slip. We cannot estimate the exact velocity of the vibration. However, the relative comparison of frequency vs conductance state of the QCM in slip and no-slip conditions is now shared in the updated figure 1. **See below updated figure 1 schematic.***

-Fig 2: (b-d) could be moved to SI since they are merely a control. However, e is confusing (why not a bar diagram) and the color coding in (f-g) makes it impossible to correlate CA with the observations.

We agree with the reviewer's comment and now we have moved the figure f-g in the supporting information. The updated figure 2 is shared below where we have also ensured that the color coding is consistent in all subfigures. Note that we have also changed the contact angle subfigure (figure e in our 1st submission) to bar diagram.

-Fig 4 e suggests no particular correlation between CA and slip magnitude, but rather only a clear signature of flow rate. This deserves more comments discussion since it is not expected.

We apologize for our poor representation of this important subfigure. In fact, we do see a relationship between slip and contact angle. We show this by connecting lines for individual flow rates. This figure describes 2 elements: 1) with flow rate slip increases (as indicated by reviewer in this question) and 2) slip increase with contact angle (as shown now by connecting lines). Please note that due to the closeness of slip values for MPTS, plasma and untreated samples (hydrophilic category surfaces) the scale of the figure can resolve large shift in the slip only for high flow rates and at higher contact angles. However, with contact angle we see increase in values of slip as a general trend for all flow rates.

With connecting lines, the reader is also left wondering about minute fall in slip for MPTS compared to plasma (for some flow rates, not all). This is due to the fact that oxygen plasma treatment is short lived and during the flow of the fluid (during experiment) some of the -OH bonds may not survive for long and as a result it becomes slightly more hydrophobic than what its contact angle suggests. Nevertheless, there are significant number of -OH bonds to consider it hydrophilic compared to untreated surface as suggested by slip values, contact angle, FTIR and AFM measurements.

Based on this we have also added connecting lines in figure 4f by ourselves for consistent description of our findings (not asked by reviewer).

Additionally, we have added some discussion on this in the manuscript. **See below added text and changed figure 4e and 4f:**

Note that the slip for MPTS is less as compared to oxygen plasma whose contact angle is lower than MPTS. This is due to the fact that oxygen plasma treatment is short lived (as also mentioned earlier) and during the flow of the fluid (during experiment) some of the -OH bonds may not be present on the surface for longer duration and as a result it becomes slightly more hydrophobic than what its contact angle suggests. Nevertheless, there are significant number of -OH bonds to consider it hydrophilic compared to untreated surface as suggested by slip values, contact angle, FTIR and AFM measurements.

Reviewer #2 (Remarks to the Author):

Slip at the solid-liquid interface is an interesting topic in many research areas. This work presents a novel experiment approach which incorporates the QCM (quartz crystal microbalance) sensor into a micro-channel of fluids. Meanwhile, a relationship between the resonant frequency shift and energy dissipation ratio of the sensor with the slip length is established. The predictions of the model are validated by measurements by using functionalized sensors with different contact angles in a wide range of flow rate. The main findings of the study is the contribution of an inertial component to the slip length at higher flow rates. There are a few points need to be clarified in revision before it is accepted for publication.

We thank the reviewer for appreciating our work and for suggestion our work for publication. Below are our actions and clarifications on questions asked by the reviewer.

1. Did you measure the pressure difference between the inlet and outlet in the microfluidic experiments? Based on fluid mechanics of Newtonian fluids, the effective slip length can be calculated from the test results of flow rate and pressure drop. The slip length estimated with the QCM measurement data and Eq. (4) should be compared with the effective slip length.

We thank the reviewer for this comment and it is a very interesting point to measure inlet and outlet pressure. To answer precisely, we did not measure the difference between the inlet and outlet pressure.

This is because the magnitude of the slip length for a smooth surface is highly varied with studies ranging from no-slip to tens of nanometers as described by recent work of Sanchez et al. [Advanced Materials interfaces, 9, 2101641 (2022)]. This is attributed to the fact that the pressure drop caused by liquid slip on a smooth surface is too small to be accurately measured with conventional pressure drop instruments.

Based on the mechanics of fluid, for a fully developed laminar flow between two infinite parallel plates, the pressure drop can be estimated as

$$\Delta P = \frac{4\mu L Q}{H^3} \left(\frac{1}{3} + \frac{b}{b+H} \right)^{-1}$$

where H , Q , b , are the distance between the plates, the flow rate, and slip length, respectively. Yes, using the above pressure drop versus flow rate method, the effective slip length can be estimated. An important implication of this is that substantial pressure drop is only possible when the slip length is of the same order of magnitude as the characteristic length of the system. This is why measuring liquid slip is extremely difficult on smooth surfaces.

*For instance, the calculated pressure drop difference using the above equation with and without liquid slip **length close to 50 nm** is on the order of **10^{-5} to 10^{-6} Pa** which is unlikely to be measured using the conventional low-pressure gauge incorporated into a microfluidic system such as ours. As an alternative way, researchers try to use*

fluorescent particle imaging with micro-PIV system or other laser techniques to determine boundary liquid slips on smooth surfaces.

On the contrary, achieving large liquid slip lengths and high-pressure drop is possible by implementing surface roughness by surface modification. The so-called effective slip length values on roughened surfaces were reported to be on the order of several micrometers to tens of micrometers, thus can be measured by using conventional pressure drop methods. Overall, the effective slip length on roughened surface can be measured with conventional pressure drop methods, however, the boundary liquid slip needs much more sensitive and new techniques to measure as demonstrated by our work.

2. The dissipated energy in the microfluidic transforms into heat and makes the temperature of fluid rise. Did you account the temperature effect in derivation of Eqs.(3) and (4)?

We thank the reviewer for this comment. We would like to share that we have measured change in temperature in our experiments using the in-built temperature sensor in the QCM measurement unit of OpenQCM instrument. The temperature changes in the 6 different sensors with the flow rate of 30 ml/hr are observed to be less than 0.6 °C, see figure below. The base line temperature is 25 ± 1 °C. This suggests that there is less than 1.5% change in temperature (on average) at the highest flow rate tested in our experiment and therefore the temperature effects are negligible in our experiments performed to validate our analytical model.

Additionally, we would like to point out that we have used AT cut quartz crystal in our system whose frequency does not change with temperature variations of 1-2 °C around 25 °C [Sensors 11, 4474-4482 (2011)]. Therefore, in our experiments/model we can neglect the effect of temperature.

However, we would like to consider temperature variations in slip length in our future work for which a system involving obvious variations in temperature (maybe use of heater with microfluidic structures) will be considered to study temperature effects. We again thank the reviewer for this suggestion here.

Note that we have now added the above graph in supplementary information.

3. As shown in Fig.3, the slip length increases with flow rate monotonically. If flow rate is much larger than 32 ml/hr, how the slip length changes?

We thank the reviewer for this question. With increase in frequency of QCM, from mathematical perspective, the slip length computed by our model will increase as pointed by the reviewer.

The question now is how much the frequency of QCM will change upon increasing the flow rate beyond 32 ml/hr. If the frequency increases monotonically the slip length will also increase monotonically. However, we do not perform experiments beyond 32 ml/hr to avoid several effects: 1) to ensure stability of the functionalized surface; 2) to avoid harmonic noise generated by pumping fluid at high flow rates that may result in change in the natural frequency of our QCM and 3) to some extent limit friction (which is higher for higher flow rates) that may also generate unwanted noise due to temperature change. In future we will work on the multi-mode vibrating solids where we aim to eliminate noise which will reduce the apparent experimental artifacts (the one aforementioned) at higher flow rates.

4. The authors assume that slip occurs between the first liquid layer and the solid surface. Is this assumption always true? For water, there are experiment measurements showing that the water molecule nearest to the solid surface are solid-like.

To provide clarification for this point, at the first instance we would like to mention that we have considered 3 cases to elucidate the slip behavior:

- 1) vibrating solid-liquid interface boundary condition without slip,*
- 2) vibrating solid-liquid interface boundary condition with slip, and*
- 3) vibrating solid-liquid interface boundary condition with slip including inertia force.*

Note that we have mentioned these in our manuscript.

*So, in our third case, we add the effect of the first liquid layer and solid surface to our assumption by modeling the inertial force which is considered here as the length of first water layer on the solid surface. This length of first water layer contributes to the momentum/energy transfer at the liquid/solid interface. Moreover, as referee mentioned, generally the behavior of water molecules near the solid surface at nanoscale is different than the bulk water. This is because the liquid layer near a solid surface forms an interfacial layer where the molecular structure of liquid is different from that of the bulk structure. This is true for all solid-liquid systems. Therefore, in this case, we assume the **inertia of the liquid layer** near the interface has a **non-negligible contribution** to the friction force (more solid like) which is the hallmark of our model to calculate the slip length. To summarize, we consider the **contribution of water molecules in the first layer** to **include** effects from the interfacial region, where the properties and structures of the liquid differ from those in the bulk liquid.*

For water on the atomically smooth surface, according to the literature [Nature 379, 219–225 (1996), Nature Communications 7, 12164 (2016), Nature Communications 10, 2606 (2019)] the interfacial layer is about one monolayer thick; however, this thickness may vary depending on the surface conditions [Nature Communications 10, 2606 (2019)]. So, through our modelling approach which leads to the equation 4 of the manuscript, we exactly cover the point raised by the respected reviewer.

Furthermore, our proposed equation can pave the way to consider the effect of interfacial region on the slip in different conditions including different surfaces (e.g. hydrophobic/hydrophilic) and different liquids (e.g. with different ion concentrations).

5. References [3] and [15] missed information.

We apologize for the typesetting error here. We have now fixed the missing information in the references [3] and [15].

REVIEWERS' COMMENTS

Reviewer #1 (Remarks to the Author):

The authors have extensively revised their manuscript taking into account my comments. They have addressed the main concerns I had with respect to independent validation of the model they develop. I still have some minor concerns with experimental aspects and comparability, but I feel the paper is now ready to be discussed by the scientific community at large.

Reviewer #2 (Remarks to the Author):

My concerns have been addressed in the revised manuscript. I have no further questions.